SciPost Physics

Submission

# Multiband quasi-particle interference in the topological insulator $Cu_xBi_2Te_3$.

E. van Heumen[1*], G.A.R. van Dalum[2], J. Kaas[1], N. de Jong[1], J. Oen[1], Y.K. Huang[1], A.K. Mitchell[3], L. Fritz[2], M.S. Golden[1]

**1** van der Waals - Zeeman Institute, University of Amsterdam, Sciencepark 904, 1098 XH Amsterdam, the Netherlands
**2** Institute for Theoretical Physics, Utrecht University, Princetonplein 5, 3584 CE Utrecht, Netherlands
**3** School of Physics, University College Dublin, Dublin 4, Ireland
* e.vanheumen@uva.nl

January 29, 2019

## Abstract

We present angle resolved photoemission experiments and scanning tunneling spectroscopy results on the doped topological insulator $Cu_{0.2}Bi_2Te_3$. Quasi-particle interference (QPI) measurements, based on high resolution conductance maps of the local density of states show that there are three distinct energy windows for quasi-particle scattering. Using a model Hamiltonian for this system two new scattering channels are identified: the first between the surface states and the conduction band and the second between conduction band states. We also observe that the real space density modulation has a predominant three-fold symmetry, which rules out a simple, isotropic impurity potential. We obtain agreement between experiment and theory by considering a modified scattering potential that is consistent with having mostly Bi-Te anti-site defects as scatterers.

# 1 Introduction

The discovery of topological insulators (TIs) [1–3] has opened a new route to low energy electronic devices, whereby the surface states taking part in the transport are undisturbed by non-magnetic impurity scattering. Soon after the discovery of the topological phase in $BiSb_x$ [4], a new family of TIs was predicted and discovered: $Bi_2Se_3$, $Bi_2Te_3$ and $Sb_2Te_3$ [5,6]. It became apparent that higher order terms in the $\mathbf{k} \cdot \mathbf{p}$ expansion are relevant for describing the low energy electronic states, leading to a warping of the Dirac cone away from the Dirac point [7,8]. Such warping has been observed in angle-resolved photoemission spectroscopy (ARPES) experiments [9] and has been shown to have an important impact on local density of states (LDOS) oscillations observed in scanning tunneling microscopy/spectroscopy (STM/S) [10–21].

In this article we use STS, supplemented by ARPES and theoretical modeling, to find different regimes of carrier scattering in Cu doped $Bi_2Te_3$. The STS data indicates three different regimes of scattering with dominant $\mathbf{q}$-vectors in the direction of the reciprocal lattice vectors ($\Gamma$ - $M$). A direct comparison between the STS and ARPES data suggests that the first regime arises from scattering within the surface state band, while a transition to a new scattering regime is observed when the surface states start to overlap with the conduction band. We use the ARPES data to construct a $\mathbf{k} \cdot \mathbf{p}$-Hamiltonian that allows us to calculate the scattering vectors within a single impurity T-matrix formalism. The best agreement between theory and experiment is achieved when an impurity potential is associated with the Bi sites in the lattice model. This suggests that the LDOS oscillations are dominantly arising from $Bi_{Te}$ anti-site defects [22].

# 2 Experimental methods.

Single crystals of $Cu_{0.2}Bi_2Te_3$ were grown using a Bridgman technique. The elements were sealed in a quartz tube, heated to 850 °C and held there for 10 hours. This was followed by a slow cool (1 °C/hour) to 600 °C, the samples being held at this temperature for a further 60 hours. ARPES experiments were performed with a lab-based system equipped with a high intensity helium discharge source. Angular and energy resolution were set to 0.008 Å$^{-1}$ and 6 meV. ARPES data was taken at 18 K. UHV STM measurements were performed at 4.2 K, using Pt/Ir tips to achieve a tunneling current of 0.5 nA with junction resistances between 0.1-0.5 GΩ. Conductance maps, $g(\mathbf{r}, E)$, have been recorded over $60 \times 60$ nm$^2$ areas using a modulation frequency of 969 Hz with a bias modulation amplitude of 5 meV. In all experiments samples are cleaved *in situ* at room temperature.

# 3 Quasi particle interference experiments.

A powerful probe of quasi-particle scattering on TI surfaces is provided by conductance mapping in spectroscopic-imaging STM [10,11,13,23–25]. In Fig. 1(a,b) we present representative conductance maps at two different energies. There are clear standing wave-like patterns in $g(\mathbf{r}, E = 100$ meV$)$ originating at impurity sites, whose characteristic wavelength ($\approx 30$ Å) can be picked up in the Fourier Transform (FT), $\rho(\mathbf{q}, \omega)$, shown in the inset. $\rho(\mathbf{q}, \omega)$ at $E =$ -175 meV also shows a similar FT pattern, whereby the lobes arranged with six-fold symmetry are clearly closer to the origin, making the associated longer wavelength patterns difficult to spot with the naked eye in the raw LDOS data. Many such LDOS maps have been measured, generating $\rho(\mathbf{q}, \omega)$ images spanning the bias interval -350 meV to 250 meV in 12.5 meV steps. At the energies where peaks are observed in $\rho(\mathbf{q}, \omega)$, they always appear along the same directions in $q$-space ($\Gamma \rightarrow M$). We determine the $q$-vectors for the six quasi-particle interference (QPI) peaks and plot the average $q$-vector in Fig. 1c. Error bars are determined from the spread in the peak positions in the six directions. We can identify three different regimes in the QPI patterns: (I) -300 meV to -175 meV, (II) -125 meV to 125 meV and (III) $\geq 200$ meV. To understand the origin of these different regimes, we will compare the measured dI/dV to ARPES measurements, which we will now discuss.

# 4 Angle resolved photoemission experiments

Figure 2a shows the Fermi surface (FS) of $Cu_{0.2}Bi_2Te_3$. The surface state dispersion displays the hexagonal deviation (warping) from a perfect Dirac cone similar to undoped $Bi_2Se_3$ [7,8]. An energy/momentum intensity map, $I(E,k)$, along the $\Gamma \rightarrow M$ direction is shown in Fig. 2b, while $I(E,k)$ along $\Gamma \rightarrow K$ is shown in Fig. 2d. From the $I(E,k)$ maps we can extract the surface state dispersion by making fits with Lorentzian line-shapes to the momentum distribution curves. The peak positions thus obtained are shown in panels 2b,d by blue squares. These peak positions allow us to obtain an estimate of the Dirac point energy $E_D \approx$ -385±10 meV. The data presented in Fig. 2 was taken in a lab-based system with a 'low flux' source [26]. Before recording the ARPES map, 2 hours were spent aligning the sample, while the light spot illuminated the entire sample surface area ($\approx 4$ mm$^2$). We can therefore safely assume that downward band bending resulting from the exposure to residual gases combined with UV radiation has saturated [26]. Consequently, the data presented in Fig. 2 can be used as reference for the STM conductance maps, which were recorded on a similar crystal.

# 5 Connecting QPI and ARPES experiments.

The different regions observed in $\rho(\mathbf{q}, \omega)$ in Fig. 1(c) can be easily connected to features of the bandstructure observed in the ARPES experiments shown in Fig. 2. Region I spans the energy window between the top of the valence band (VB) and the conduction band (CB) bottom. In this special energy interval only quasi-particle scattering between surface states (SS-SS) is possible, facilitated, where relevant, by the Dirac cone warping [7]. For energies close to the bottom of the conduction band the QPI patterns are suppressed and then a new QPI regime is encountered (dubbed region II), matching an energy window in which both SS

and CB states are available for scattering. The largest $q$-vector in region I observed in the STM experiments is approximately 0.2 $\pi/a$, and can be identified with a wave vector of 0.21 $\pi/a$ in the ARPES data connecting opposite sides of the surface state dispersion at the same binding energy. Likewise, the shortest $q$-vector in region II from STM is approximately 0.15 $\pi/a$ and matches well with the observed distance between the bottom of the conduction band and the surface state dispersion from ARPES (0.13 $\pi/a$). In the following, we will show that region II is indeed dominated by scattering between surface states and the bulk conduction band (SS-CB). Finally, the dispersion observed at energies above the second jump in the $q$-vector around +150 meV (indicated in Fig. 1(c) as region III) could be due to scattering within the conduction band.

## 6 Simple impurity scattering modeling of QPI.

To underpin the assignment of the different quasi-particle interference regions, and in particular the multi-band scattering underlying region II, we utilize an extension of an electronic structure model based upon the work of Ref. [8] and applied to simulate QPI in Refs. [27,28]. The electronic structure of the tetradymite topological insulators can be described by the Hamiltonian

$$
\begin{aligned}
H = \varepsilon(\mathbf{k})\mathbb{I}_4 + \mathcal{M}(\mathbf{k})\Gamma_5 + \\
+ \mathcal{B}(k_z)\Gamma_4 k_z + \mathcal{A}(k_{||})(k_y\Gamma_1 - k_x\Gamma_2) + \\
+ R_1\Gamma_3(k_x^3 - 3k_x k_y^2) + R_2\Gamma_4(3k_y k_x^2 - k_y^3)
\end{aligned}
\tag{1}
$$

where $\mathbb{I}_4$ is the identity and $\Gamma_{1-5}$ are Dirac matrices satisfying the Clifford algebra $\{\Gamma_i, \Gamma_j\} = 2\delta_{i,j}$ and the functions $\varepsilon(\mathbf{k}) = C_0 + C_1 k_z^2 + C_2 k_{||}^2$, $\mathcal{M} = M_0 + M_1 k_z^2 + M_2 k_{||}^2$, $\mathcal{B} = B_0 + B_2 k_z^2$ and $\mathcal{A} = A_0 + A_2 k_{||}^2$. For systems such as $Bi_2Se_3$ and $Bi_2Te_3$, the last two terms in Eq. 1 break the in-plane symmetry, producing the Dirac cone warping.

Starting from the model Hamiltonian, Eq. (1), we calculate the LDOS using the method outlined in [2,29]. The Hamiltonian is compactified on a cubic lattice with periodic boundary conditions in the $x-y$ plane and open boundary conditions in the $z$-direction, thus describing a semi-infinite slab as a function of in-plane momentum $\mathbf{k}_{||}$ and interlayer hopping parameters. The surface Green's function in the absence of impurities is calculated employing the exponentially fast iterative scheme proposed in [30], which is in turn used to calculate the spectral function $\rho(\mathbf{k}_{||}, \omega)$. The parameters from the model Hamiltonian are fixed by matching $\rho(\mathbf{k}_{||}, \omega)$ with the ARPES data from Fig. 2 [31]. From this spectral function QPI is simulated by introducing a single, momentum-independent local scattering potential $V$ on the surface of the system and using standard T-matrix formalism to find the corresponding LDOS function. (Note: an effective surface Hamiltonian, $H_{2D} = E(k) + v_k(k_x\sigma_y - k_y\sigma_x) + \frac{\lambda}{2}(k_+^3 + k_-^3)\sigma_z$ with $v_k = v(1 + \alpha k^2)$, where $v = 3.1$ eVÅ, $\alpha = 5.8$ eVÅ$^2$ and $\lambda = 50$ eVÅ$^3$ obtained through fits to the ARPES data in Fig. 2b and 2d, has been used in the literature to study QPI patterns in STS experiments [27,28] arising from SS↔SS scattering.)

To make meaningful comparisons between theory and experiment, one must also determine the form of the impurity scattering potential. The simplest form is an isotropic impurity potential, coupling equally to all spin and orbital degrees of freedom in the lattice unit cell. This was used in previous studies [27,28]. However, in general one must account for the specific way in which the impurity or defect couples to the underlying lattice degrees of freedom. As

shown below, this can make a significant difference to the resulting scattering observed in both real and momentum space. With an isotropic impurity potential, we find agreement with the experimental results *only* above the energy where the hexagonal energy contours change from convex to concave ($E > $ -150 meV). For energies below $E \approx$ -150 meV, the orientation of the FT-LDOS patterns rotates by 30 degrees in disagreement with experiments (the 30 degree rotation with increasing energy underlies the apparent discrepancy between results presented in Ref. [27] and Ref. [28]).

A hint towards resolving this problem comes from a comparison of the experimental real space imaging of LDOS oscillations around defects, which can be seen in Fig. 1a and 1b, and the theoretical surface LDOS oscillations. In fig. 3a we show the real space density modulation corresponding to simple, isotropic impurity scattering. As expected, the density modulation displays a clear six-fold symmetry arising from the hexagonal warping introduced in Eq. 1. The bottom row of Fig. 3 (panels c-e) show enlarged images of the typical defect structures that are seen in our STM experiments and they are clearly dominated by three-fold symmetry (Fig. 1a and [12, 16]). To resolve the discrepancy between the experimental and theoretical real space density modulations it is important to understand the origin of the defects. For the $Bi_2Se_3$ and $Bi_2Te_3$ materials the origin of the defects is slightly different. In Ref. [32] it was pointed out that the defect chemistry in $Bi_2Te_3$ results mostly in the formation of anti-site defects where Bi atoms sit on Te sites and vice versa.

To include this experimental observation in the theoretical model, we exploit the generalised form of the impurity/defect potential. To most faithfully capture the experimental results, we find that the impurity potential should not be isotropic, but rather it should be associated only with the Bi lattice sites. The best results are obtained for a non-magnetic impurity in the weak-scattering ("Born") limit. With this specific impurity matrix, the numerical LDOS oscillations have the same rotational symmetry as the STS data as can be seen in Fig. 3b. Although the comparison between real space experimental and theoretical density modulations now looks very promising, the impact on the momentum space images is less pronounced. Fig. 4a shows the comparison between experimental (left) and theoretical (right) momentum resolved LDOS images. The experimental QPI pattern is clearly peaked along the $\Gamma \to M$ direction, while the model indicates the largest scattering along $\Gamma \to K$. The main difference between isotropic and non-isotropic scattering lies in the high momentum 'streaks'. These are much weaker or absent in the isotropic case. Similar streaks can be seen in the experimental data. We reiterate that the real-space images provide the best indication that a simple isotropic impurity potential is not sufficient.

Returning to the shift in momentum that is observed in experiments between regions I and II in fig. 1, we also find from the theoretical modeling that there is a significant reduction in the overall LDOS oscillation amplitude for energies close to the CB bottom. Figure 4b shows the resulting QPI pattern in between region I and II. Including the bulk CB in the calculation has the effect of smearing out the QPI intensity as a result of the large LDOS associated with the bottom of the CB. This observation, in combination with the comparison to the ARPES experiments discussed above, suggests that the inclusion of scattering between the SS and the CB is vital. We note that in our calculation the exact energy where this occurs ($E_{th}$ = -50 meV) is somewhat higher than that obtained in experiment ($E_{exp} \approx$ -140 meV) as a result of small errors in the estimated model parameters. In region II, the QPI intensity returns as the velocity of the bulk CB increases (Fig. 4c). The jump in extracted momentum values matches well with the momentum difference between the SS and CB states (indicated with a yellow arrow in Fig. 4c).

# 7 Conclusion

To conclude, we have analyzed the quasi-particle interference patterns on the surface of Cu-doped $Bi_2Te_3$. We find that the scattering patterns are indicative of three regimes: surface state to surface state scattering, surface state to conduction band scattering and finally bulk-bulk scattering. We note that previous models used for simulating QPI interference on tetradymite surfaces do not accurately reproduce the experimental observations. The symmetry of the real space density modulations displays a threefold symmetry, rather than the expected six-fold symmetry. We reproduce this lower symmetry by considering a modified impurity potential, which we suggest arises from the presence of mostly Bi-Te anti-site defects in $Bi_2Te_3$. Our modified impurity potential does improve the correspondence between experimental and theoretical momentum resolved QPI patterns.

**Funding information**  EvH and MSG are supported by the foundation for Fundamental Research on Matter (FOM) program Topological Insulators, while LF and GvD are supported through the D-ITP consortium, both part of the Netherlands Organization for Scientific Research (NWO) that is funded by the Dutch Ministry of Education, Culture and Science (OCW).

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

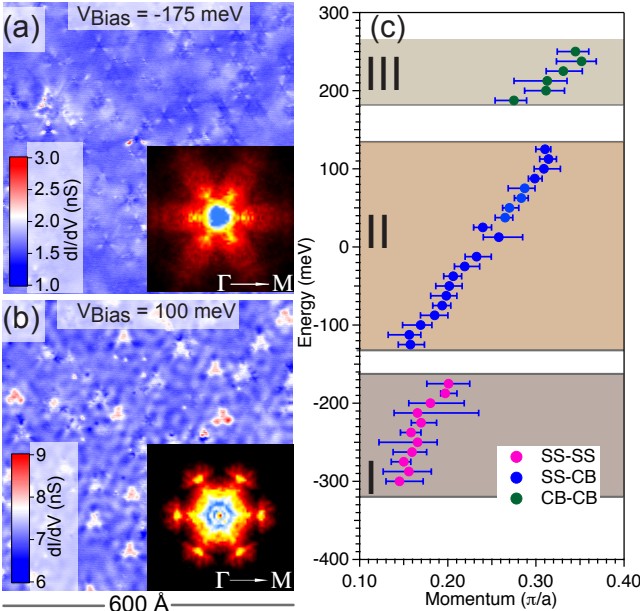

Figure 1: (Color online) (a,b): LDOS maps measured at $V_{\text{Bias}}$ indicated. Insets show FT-LDOS, which have been three-fold symmetrized. (c): Dispersion of QPI $q$-vectors along the $\Gamma \rightarrow M$ direction obtained from the FT-LDOS. Three different scattering regimes are indicated: I, SS↔SS scattering due to Dirac cone warping; II, SS↔CB and III: CB intraband scattering.

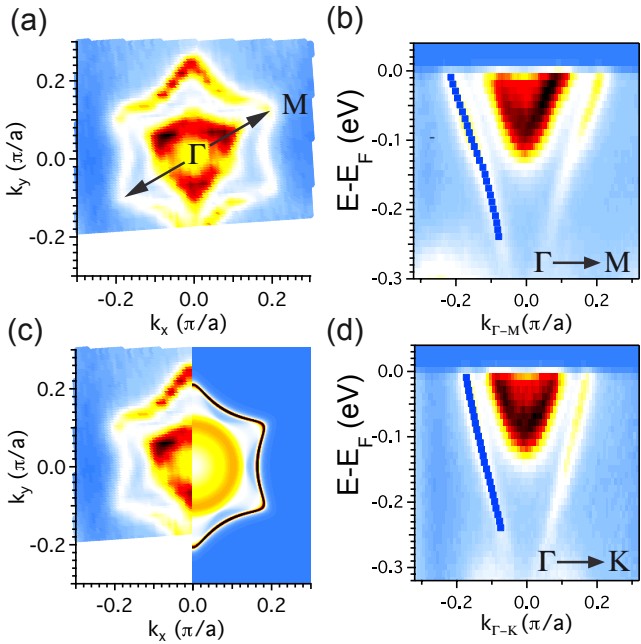

Figure 2: (Color online) (a): Fermi surface taken with $\hbar\nu = 21.2$ eV at T = 18 K. The high symmetry direction $\Gamma \to M$ is indicated. (b,d): $E, k$-intensity maps along the two high symmetry directions $\Gamma \to M$ and $\Gamma \to K$, respectively. The blue squares indicate the surface state dispersion obtained from fits of momentum distribution curves. It is clearly seen that the dispersion bends outwards in the $\Gamma \to M$ direction. (c): comparison of the ARPES data with the Fermi surface calculated from the model Hamiltonian, Eq. 1.

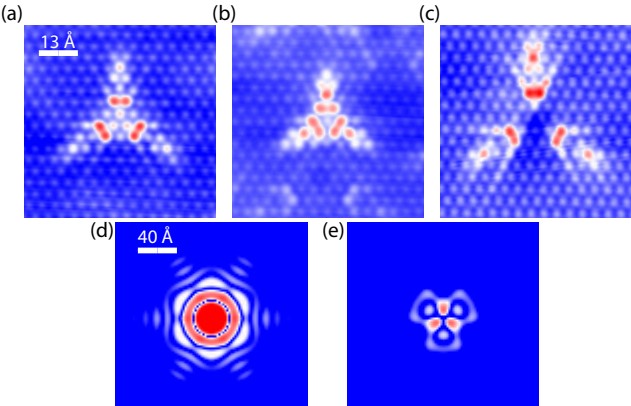

Figure 3: (Color online) (a): simulated real space density modulation for a simple, isotropic impurity potential. The impurity is extended and shows six-fold symmetry. (b): real space density modulation for a modified impurity potential. The impurity is more compact and has lower, three-fold symmetry. (c)-(e): Experimental defect images. Practically all defects observed in experiments display a three-fold symmetry.

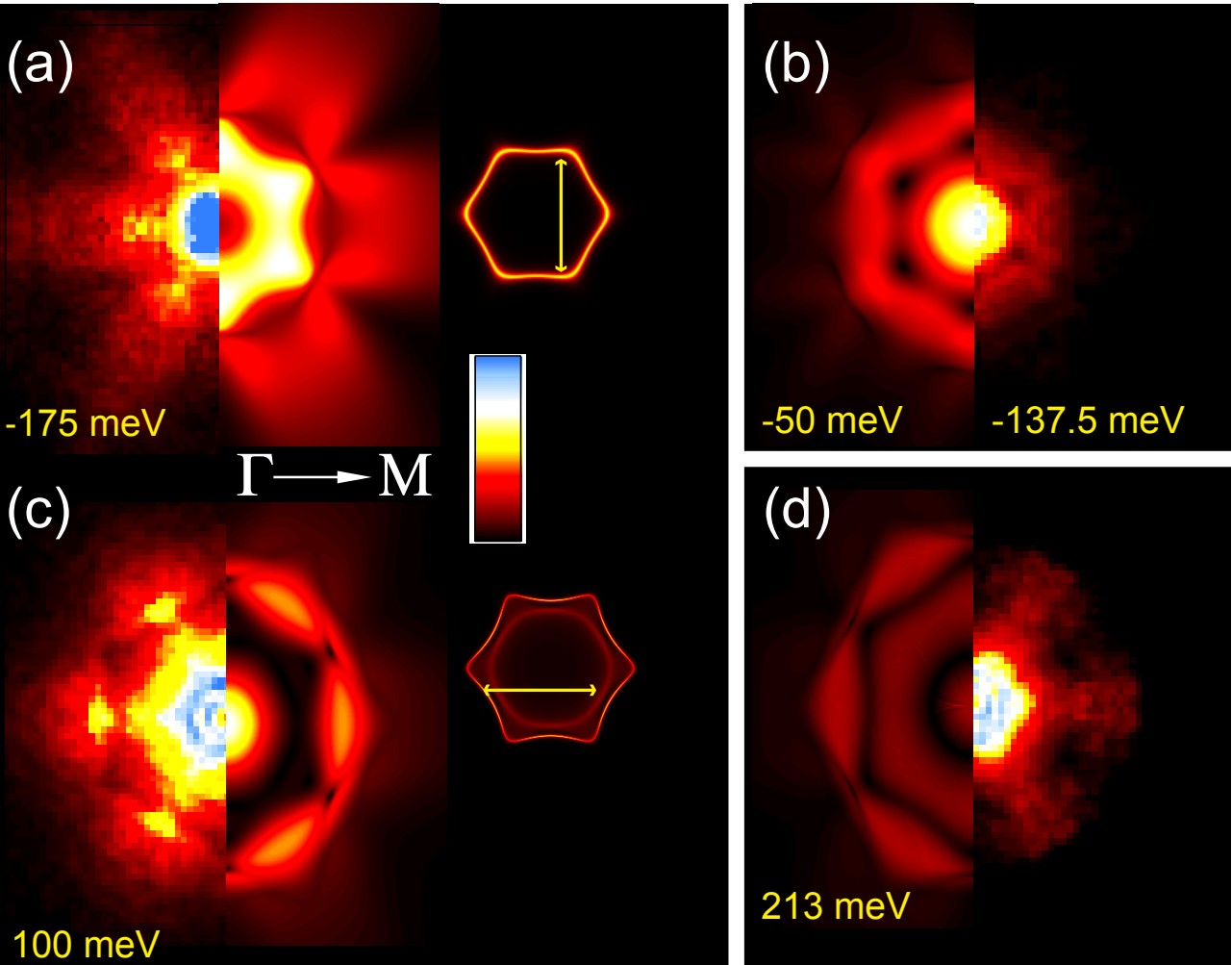

Figure 4: (Color online)(a,c): Comparison between $\rho_{exp}(\mathbf{q}, \omega)$ (left) and $\rho_{calc}(\mathbf{q}, \omega)$ (right) for energies of -175 meV (a) and 100 meV (c). Also shown are the relevant constant energy contours of the spectral functions with arrows indicating the dominant contributions to the scattering. (b): FT-LDOS pattern for -50 meV (simulation, left) and -137.5 meV (experiment, right). Both the experiment and the calculation show a suppression of the overall intensity compared to the patterns in panels (a,c). (d): FT-LDOS pattern for 213 meV well above the conduction band bottom.