# Peer review of "Multiband quasiparticle interference in the topological insulator Cu_(x)Bi_(2)Te_(3)"

_SciPost Physics_

## Round 3 · Referee Report · Anonymous (Referee 1) · 2019-5-28

Strengths

1- Detailed energy dependence of QPI patterns 2- Possibly interesting effects from impurity model in the theoretical analysis

Weaknesses

1- experimental results similar to those already in literature 2- theoretical model presented not very clearly 3- probably no realistic description of bulk states

Report

The authors show quasiparticle interference (QPI) experiments together with angle resolved photoelectron emission spectroscopy (ARPES) and numerical simulations on Cu-doped Bi2Te3. Depending on the energy where the scanning tunneling spectroscopy (STS) maps are recorded, they classify three regions where the QPI patterns are proposed to be generated by different scattering channels: an intraband surface-state scattering region between -300 and -150 meV, a surface-to-bulk scattering region between -130 and +130 meV and a bulk intraband scattering region above 180meV. They use a model Hamiltonian proposed by Liu et al. [8] and a surface Green's function technique to obtain the spectral functions and simulate QPI maps. They conclude that it is necessary to introduce a non-isotropic scattering potential to match (to some extent) theoretical and experimental findings.

Reading the manuscript I was wondering what could be the main message of this work: several experimental QPI and ARPES studies of Bi2Te3 can be found in the literature, but it seems that the influence of Cu doping was not in the focus here, the spectra look similar. Therefore, I concluded that the explicit consideration of the non-isotropic scattering potential could be the new point
here.

Focusing on this point, I have to admit that it became not clear to me how exactly this non-isotropy was included. The authors start from the Hamiltonian (1) and use probably an impurity model like in Ref.[29] of the manuscript, but how the impurity matrix is adjusted and what are the induced changes in the QPI patterns is not clear: the authors mention that the appearance of streaks depends on this but obviously other features (like the rotation of the inner hexagon) are still at variance with the experiments.

If I take e.g. Phys. Rev. B 88, 161407(R) (2013) where similar experiments are compared to density functional theory (DFT) calculations of Bi2Te3, I see that the shape of the bulk valence states is quite important to explain the experiments. Of course DFT also has some shortcomings, but compared to GW calculations of these materials [e.g. Phys. Rev. B 88, 045206(R) (2013)] it seems that many features are robust. In contrast Hamiltonian (1) was designed to describe the surface state dispersion, but for the bulk states I doubt that reliable dispersions can be obtained. Maybe some features can be simulated by changing the scattering potential but over a wide energy range this might be problematic.

Summarizing, I'm not convinced that the manuscript enhances our understanding of quasiparticle scattering on Cu doped Bi2Te3. Maybe I missed some points of the authors, the presentation is not always systematic and clear, but then I'd suggest to express this more clearly.

Requested changes

1- Analyse difference to literature results of Bi2Te3 (QPI/ARPES) 2- Explain impurity model more clearly and how it affects the QPI patterns 3- Reconsider the model description of the bulk states and influence on the scattering channels 4- Correct labelling of figure panels in figure 3

---

## Round 3 · Referee Report · Anonymous (Referee 2) · 2019-6-6

Strengths

1 - good data
2 - comparison of different techniques and theoretical models

Weaknesses

1 - little novelty
2 - not clearly written
3 - some citations missing

Report

This paper presents nice data, and it makes important comparisons between theoretical models, ARPES, and STM. Nothing seems very new here, there are many many papers on Be2Te3 doing exactly the same. (The authors use Cu doped materials, but this is not really addresses, nor is it shown that the sample really has Cu in it.) In general, the paper seems written in a haste, and does not live up to the quality of the data.

I am, for example, confused about the Cu. How do the authors know that it is in there? If there is really x=0.2, as written in the abstract, would one not expect huge change in the properties? There is literature about this.

Often it is very unclear what is novel in the present paper. The authors write e.g. "We note that previous models used for simulating QPI interference on tetradymite surfaces do not accurately reproduce the experimental observations.” I am not an expert, but the papers cited by the authors claim that they do accurately reproduce the experimental observations. And they look fine to me. So if the authors disagree, they need to explain why. The also write “extension…” (section 6, line 2), but don’t elaborate. Why is new, e.g. with respect to this paper (which they do not cite): https://iopscience.iop.org/article/10.1088/1674-1056/22/6/067301/pdf

Further, the authors stress the importance of considering anisotropic scattering potentials. But that is already discussed in
http://citeseerx.ist.psu.edu/viewdoc/download?doi=10.1.1.727.8548&rep=rep1&type=pdf , as well as in the paper
http://dx.doi.org/10.1038/nphys2108 (mainly in the SI), which is also not cited by the authors.

I admit I do not really know the standards of SciPost in terms of novelty and quality, but looking at what is already published, I tend to say that the present paper does not live up to it.

As a detail, the authors might want to check capitalization and spelling, especially in the references, and the captions.

---

## Editorial Decision

awaiting_resubmission